# The Performance of CR180IF and DP600 Laser Welded Steel Sheets under Different Strain Rates

**DOI:** 10.3390/ma14061553

**Published:** 2021-03-22

**Authors:** Mária Mihaliková, Kristína Zgodavová, Peter Bober, Anna Špegárová

**Affiliations:** 1Faculty of Materials, Metallurgy and Recycling, Technical University of Košice, Letná 9, 042 00 Košice, Slovakia; maria.mihalikova@tuke.sk (M.M.); anna.spegarova@tuke.sk (A.Š.); 2Faculty of Electrical Engineering and Informatics, Technical University of Košice, Letná 9, 042 00 Košice, Slovakia

**Keywords:** CR180IF steel, DP600 steel, dissimilar laser welding, static properties, dynamic properties, microstructure, microhardness, welded joint roughness

## Abstract

The presented research background is a car body manufacturer’s request to test the car body’s components welded from dissimilar steel sheets. In view of the vehicle crew’s protection, it is necessary to study the static and dynamic behavior of welded steels. Therefore, the influence of laser welding on the mechanical and dynamical properties, microstructure, microhardness, and welded joint surface roughness of interstitial free CR180IF and dual-phase DP600 steels were investigated. Static tensile tests were carried out by using testing machine Zwick 1387, and dynamic test used rotary hammer machine RSO. Sheet steel was tested at different strain rates ranging from 10^−3^ to 10^3^ s^−1^. The laser welds’ microstructure and microhardness were evaluated in the base metal, heat-affected zone, and fusion zone. The comprehensive analysis also included chemical analysis, fracture surface analysis, and roughness measurement. The research results showed that the strain rate had an influence on the mechanical properties of base materials and welded joints. The dynamic loading increases the yield stress more than the ultimate tensile strength for the monitored steels, while the most significant increase was recorded for the welded material.

## 1. Introduction

The automotive industry has a significant impact on people’s daily lives worldwide and affects their safety [1]. Steel is the first choice for most manufacturers, although it is now possible to produce car bodies from a variety of materials, from traditional steel through composite materials to plastics. Thanks to improvements in the steel industry, modern steel has higher strength and lower weight. The main factors influencing manufacturers in choosing steel for car bodies are thermal, chemical, mechanical resistance, manufacturing efficiency, and durability. Steel sheets used for body construction have a natural ability to absorb energy in the event of an impact and ensure higher safety [2]. Moreover, the sheets have good weldability and ductility.

The individual parts of the car body have long been joined by welding. However, laser welding is gradually replacing traditional welding methods as inert tungsten gas (TIG) welding, inert metal gas (MIG) welding, and spot-welding [3] with many advantages and can significantly reduce production costs while increasing production efficiency and quality. On the other hand, laser welding affects the static and dynamic properties of welded sheets, which influence passenger safety. Therefore, thorough testing of welded metals (WM) is necessary. New testing procedures and methods play a crucial role in deciding on material selection, manufacturing processes, analysis of collected data, and taking appropriate preventive actions to improve processes [4,5]. Low-carbon and ultra-low-carbon steel sheets are frequently used materials in the automotive industry, as they can be pressed and joined into complex units at low production costs [6,7].

High-strength interstitial free (IF) steels are suitable for complex structural components of automobiles that require combined tensile ductility and deep drawing. The increase in plasticity is achieved by reducing the carbon concentration to 0.01% and less and by microalloying. Interstitial free steels have no interstitial solute atoms to strain the solid iron lattice. Liquid steel is processed to reduce C and N [8,9]. Then the steel is stabilized by small additions of Ti and Nb. These steels have low yield strength from 100 to 310 MPa [10] and good ductility [11].

Dual-phase steels (DP) belong to advanced high-strength steels (AHSS). One of the essential specifics is the contribution of these steels to safety. Automobile components made of AHSS can better absorb energy in a collision with a foreign body, thus preventing the penetration of dangerous objects and their fragments into the vehicle cabin [12,13]. Dual-phase steels are generally characterized by a structure containing a martensitic and a ferritic component. A specific heat treatment achieves this microstructure. The steel is first intercritically annealed in the range of recrystallization temperatures Ac1 and Ac3 [14], while austenite and ferrite are transformed in the stable phase [15,16]. The process follows with hardening, in which the material cools very rapidly, which results in the conversion of austenite to martensite [17,18].

The surface quality of welded steel depends mainly on the parameters and conditions of welding. The high surface roughness of welded joints could be undesirable in the further processing of sheets. Laser beam welding is now standard in many industrial areas that require precision and high manufacturing efficiency. Laser welding surpasses traditional welding techniques with speed, thin welding, strength, easy integration, contactless process, and minimal maintenance costs. Due to the small heat input, the welded joint is significantly less prone to deformation due to material expansion [19]. The disadvantages of laser welding are low-energy efficiency, strict requirements for the purity of working gases (CO_2_, nitrogen, and helium), and material thicknesses up to about 2 mm for solid-state and diode lasers [20]. However, laser welding enables the joining of two dissimilar materials, as described by several authors [21,22,23].

Although research into laser-welded materials provides much valuable information, there are still problems and issues that need to be further analyzed and discussed. It concerns gathering information on less explored material pairs and examining the dynamic properties of joints, as steel manufacturers usually perform only a series of static tensile tests on welded joints before they are subjected to full-size crash tests [24].

The presented research aims to examine not only the static properties but also the dynamic behavior of laser-welded CR180IF and DP600 steels.

## 2. Materials and Methods

The investigated materials are interstitial free CR180IF and dual-phase DP600 steels in the cold-rolled and annealed condition. The thickness of both steel sheets is 1.9 mm. The measurement of the chemical composition of both materials by atomic emission spectrometry according to ASTM E415-17:2017 [25] is in Table 1.

Intestinal free CR180IF and dual-phase DP600 steels were laser welded using a solid-state fiber laser YLR 4500 (IPG Photonics, Oxford, MA, USA). Welding parameters (Table 2) are based on industrial recommendations of experts from the First Welding Company, Inc., Bratislava, Slovakia. The test specimen consists of two sheets 250 mm × 130 mm, which were clamped and welded in a protective helium atmosphere.

The metallographic samples for microstructural analysis were cut from the welded cross-section, polished, etched with a 2% Nital solution, and examined with an optical microscope Olympus Vanox AH2-AN45 (Olympus, Tokyo, Japan), and scanning electron microscope JEOL JSM-6610LV (JEOL Ltd.,Tokyo, Japan). We tested microhardness in base metals (BM), in heat-affected zones (HAZ), and fusion zones (FZ) by Vickers method according to ISO 6507-1:2018 [26] with microhardness tester Struers Duramin 5 (Struers GmbH, Villich, Germany). The indenter load was 980.6 mN (100 g) and a holding time was 10 s. Software Ecos ver. 2.37.2 (Struers, Ballerup, Denmark) automatically evaluated microhardness HV0.1 on polished metallographic samples at room temperature. The measuring points were arranged in line perpendicular to the direction of the weld, crossing the base metal, the heat-affected zone, and the welded metal zone. The distance between indentations was 0.15 mm (Figure 1).

The welded specimens were used for a static and dynamic tensile test and fracture surface analysis. Static test was realized according to ISO 6892–1:2019 [27] by the tensile testing machine Zwick 1387 (Zwick/Roell, Ulm, Germany). The mechanical properties—yield strength, ultimate tensile strength, and elongation were measured on specimens with longitudinal laser weld showed in Figure 2. Dynamic tests were performed according to ISO 26203-2:2011 [28] with rotary hammer RSO (WPM, Leipzig, Germany). The force was measured by a force sensor Quartz Load Washer Kistler with a range of 0–35 kN. The signal was sent to a charge amplifier Kistler 5015A (Kistler Instrumente AG, Winterthur, Switzerland), and processed by the oscilloscope LeCroy Wave Ace (Teledyne Le Croy, Chestnut Ridge, NY, USA). The Scope Explorer ver. 2.25 (Teledyne Le Croy, Chestnut Ridge, NY, USA) software was used to evaluate measured data (Figure 3). The speed of the hammer ranges from 5 to 50 m/s, which corresponds to an impact energy of 1.4 to 140 kJ. Because the work required to deform the specimen up to fracture is less than 60 J, the impact velocity during the test is almost constant [29]. Finally, accurate surface topography measurements were performed using an optical profiler PLu neox 3D (Sensofar Metrology, Terrasa, Spain) (Figure 4).

The principle of measuring the roughness using a confocal microscope consisted of shifting the optical system’s focus and measuring the experimental material’s surface irregularities. This method of measurement was considered noncontact because the surface layer of the material was not damaged. According to ISO 4287:1997 [30], surface roughness is defined as the sum of surface irregularities with relatively small distances resulting from the manufacturing technology used. We evaluated the mean arithmetic deviation of the profile (Ra). The measured values were subsequently graphically processed in the form of 3D maps with advanced analysis software SensoMAP ver. Standard (Sensofar Metrology, Terrassa, Spain).

## 3. Results and Discussion

### 3.1. Microstructural Analysis

The microstructure of welded joint of CR180IF and DP600 steels is presented in Figure 5. The average weld width was about 1.5 mm; the fusion zone width was 0.75 mm, and the heat-affected zone width was 0.38 mm. Figure 6 shows the microstructure of base metals, fusion zone, and heat-affected zone of the welded steels. Large images are from Olympus Vanox AH2-AN45 osciloscope, and inset are detailed images from the scanning electron microscope JEOL JSM-6610LV. The microstructure of the base material of CR180IF steel in Figure 6a was formed only by polyhedral ferrite with relatively uneven grain size. The grains size was measured by Jeffries Planimetric Method, and the average was 25 μm. A ferrite-martensite fine-grained structure characterizes dual-phase steel in Figure 6b. This steel had martensitic formations dispersed in a ferrite matrix. The structure consisted of 80% to 90% ferrite and 10% to 20% martensite. The fusion zone in Figure 6c was mainly composed of acicular ferrite microstructure, bainite, and martensite, which had different sizes and shapes.

This observation is in line with other research findings, e.g., [23]. The dissimilar laser welding of DP steels studied in the literature [31,32,33] also formed the fusion zone with nonuniform microstructure in each part of the weld, which corresponds to the chemistry during laser welding. The heat-affected zone shown in Figure 6d was created by a mixture of fine grains of transformed ferrite, possibly bainite [34]. Due to the increased temperature at the interface of the heat-affected zone of CR180IF, the ferritic grains increased.

### 3.2. Analysis of Microhardness and Its Distribution

The microhardness distribution is shown in Figure 7. The average microhardness values HV0.1 for each material are listed in Table 3.

The microhardness HV0.1 in the fusion zone differed significantly from the microhardness of DP600 and CR180IF, although the welding parameters for these steels were the same. The microhardness distribution in Figure 7 shows the highest value of 350 ± 2 HV0.1 in the transition area between HAZ-FZ on the DP600 steel side. The high microhardness value corresponds to the microstructure in the measuring area, consisting of martensite and acicular ferrite. The microhardness is given by different chemical compositions, i.e., carbon equivalent CE, calculated according to Equation (1) [35,36] from the chemical composition shown in Table 1.
(1)$$CE=C+\frac{\mathrm{Mn}}{6}+\frac{\mathrm{Cr}+\mathrm{Mo}+V}{5}+\frac{\mathrm{Ni}+\mathrm{Cu}}{15}$$
$$CE_{\;\mathrm{CR}180\mathrm{IF}}=0.024\;CE_{\;\mathrm{DP}600}=0.269$$

In the described case, the hardness reached the highest value on the side of the DP600 material, which also had a higher CE. A similar finding is described in [37], where authors claim that fusion zone hardness increases with richer chemistries producing higher CE values.

### 3.3. Static and Dynamic Tensile Tests and Fracture Surface Analysis

The mean strain rate during the tensile test was calculated according to Equation (2):(2)$$\dot{\varepsilon }=\frac{\varepsilon }{t}=\frac{v}{L_{0}}$$
where: ε is the relative deformation, *t* is the duration of the deformation, *L*_0_ is the working length of the test specimen and $\dot{\varepsilon }$ is the rate of loading. For static loading, it is 8.33 × 10^−3^·s^−1^.

Table 4 shows the results of static and dynamic tests of base metals CR180IF, DP600, and welded material (WM). Figure 8 shows the comparison of measured values of yield stress (YS) for base metals and laser-welded material.

The strain rate affected the basic mechanical properties of tested steels. The change of properties was higher at higher strain rates (Figure 8). Parametric Equations (3) and (4) describe the dependence of strength properties on the strain rates for the tested steels in the range from 10^–3^ to 10^3^ s^–1^ [38]:(3)$$YS_{\dot{\varepsilon }}=YS_{{\dot{\varepsilon }}_{0}}+A\times \ln\left(\frac{\varepsilon }{\varepsilon_{0}}\right),$$
(4)$$UTS_{\dot{\varepsilon }}=UTS_{{\dot{\varepsilon }}_{0}}+B\times \ln\left(\frac{\varepsilon }{\varepsilon_{0}}\right),$$
where: $YS_{\dot{\varepsilon }},\;UTS_{\dot{\varepsilon }}$ are yield stress and ultimate tensile strength, respectively, at a given strain rate; $YS_{{\dot{\varepsilon }}_{0}},\;UTS_{\dot{{\dot{\varepsilon }}_{0}}}$ are yield stress and ultimate tensile strength, respectively, at a static deformation rate (10^−3^ s^−1^), and *A* and *B* are material constants, which express the sensitivity of $YS_{\dot{\varepsilon }},\;UTS_{\dot{\varepsilon }}$ on the strain rate. When *A* and *B* constants were higher, the steel was more sensitive to the strain rate. CR180IF steel had *A* = 17.6 and *B* = 19.0, DP600 steel had *A* = 19.2 and *B* = 27.6, and welded material had constant *A* = 48.0 and *B* = 47.76.

The influence of strain rate on IF and DP steel’s basic mechanical properties was studied in [39,40]. The finding was that the dynamic loading increased the yield stress more than the ultimate tensile strength for the monitored steels. Our finding was that the welded material had even greater sensitivity because the constants *A* and *B* were higher than for the based material.

The fracture surface of CR180IF steel under static loading conditions is in Figure 9. Figure 9a,b shows the fracture surface under static (8.33 × 10^−3^ s^−1^) and dynamic loading conditions (1200 s^−1^), respectively. Ductile failure with pitting morphology predominated in both fractures. After dynamic stress, the holes were smaller. As seen in Figure 9b. The dimples diminished with increasing strain rate, forming voids with the increased depth and reduced size. The consequence of this was a larger number of small voids at a strain rate of 1200 s^−1^. The fracture surface of DP600 steel under static loading is presented in Figure 10a. It indicates ductile failure. The fracture surface under dynamic conditions was formed by a pit morphology caused by transcrystalline ductile failure in Figure 10b.

Under static conditions, the fusion zone’s fracture surface had a graded character of failure in Figure 11a. In the ductile fracture, a large number of dimples with different forms and volumes were found, which is consistent with [41]. Under dynamic conditions Figure 11b, brittle fracture was visible at the fusion zone, which occurred in welded materials with no macro-plastic deformation during loading. According to [42], dimples grew by coalescing and eventually formed a continuous surface layer of the fracture, and the form of the fracture was described as dimple break up. This mechanism is also visible in Figure 11b.

### 3.4. Welded Joint Surface Roughness Analysis

Thirty measurements of surface roughness were performed for base materials and the welded joint in different parts of the sample. Figure 12 shows laser confocal microscopy 3D views of individual parts with a color resolution of the heights. The base material CR180IF had an average roughness value Ra = 5.52 µm, and the 3D map is shown in Figure 12a. Figure 12c depicts the 3D map of the base material DP600, the average value Ra = 5.13 µm. The mapped surface of the welded joint with an average value Ra = 60.24 µm is shown in Figure 12b.

The roughness of welded joint is many times greater than the roughness of base materials. According to the roughness and waviness quality criteria, it may or may not meet a specific application’s requirements. Welding surface morphology has been widely studied for a long time. The literature on laser welding of two identical materials [43] and welding of two different ones [44] states that it is necessary to examine in more detail and optimize the welding parameter to reduce the surface roughness.

## 4. Conclusions

This research presents the effect of laser-welding on the static and dynamic properties of previously unexplored pair of steels DP600 and CR180IF. The experimental results indicate that the strain rate affects the mechanical properties and fracture surface of tested steels. The results from microstructure and microhardness analysis, static and dynamic tests, fracture surface analysis, and welded joint surface roughness analysis were compared with the literature and discussed. The following conclusions were drawn:

CR180IF steel contains only a ferritic structure. The microstructure of DP600 steel consists of a fine-grained ferritic-martensite structure. The heat-affected zone forms a coarse-grained ferritic structure on the side of the CR180IF steel. The fusion zone of the microstructure consists of acicular ferrite, martensite, and bainite;The calculated carbon equivalent for CR180IF steel was 0.024 and for DP600 steel was 0.269. The microhardness HV0.1 in the fusion zone differed significantly from the microhardness of base materials. The CR180IF hardness was 102 ± 2 HV0.1 and DP600 197 ± 2 HV0.1, while the hardness of the fusion zone was 245 ± 2 HV0.1 and 350 ± 2 HV0.1 on the side of CR180IF and DP600, respectively. The highest value was measured near the DP600 base material;The yield strength of CR180IF steel increased from 186 MPa in static conditions to 389 MPa in dynamic conditions (109%). The yield strength of DP600 steel increased from 393 MPa in static conditions to 721 MPa in dynamic conditions (83%). The yield strength of laser-welded material increased from 319 MPa in static conditions to 889 MPa in dynamic conditions (178%);Laser-welded material with a hybrid structure was more sensitive to strain rate. The sensitivity coefficient *A* and *B* were 48, while the base material CR180IF has *A* = 17.6, *B* = 19.0, and DP600 steel has *A* = 19.2 and *B* = 27.6. CR180IF steel with the fine-grained ferritic structure and precipitates had a lower sensitivity to the strain rate;The result of the ductile failure analysis with pitting morphology predominated in all fractures of investigates steels;The average surface roughness value was Ra = 5.52 µm for CR180IF, for DP600 was Ra = 5.13 µm, and for weld was Ra = 60.24 µm. Such a high roughness requires optimization of the welding parameters or further surface treatment.

## Figures and Tables

**Figure 1 materials-14-01553-f001:**
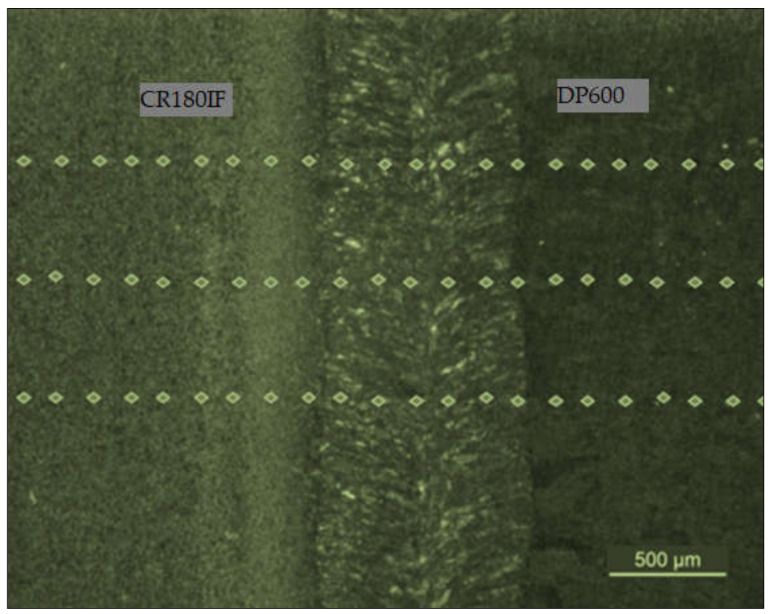
The indentation lines after surface microhardness HV0.1 measurement.

**Figure 2 materials-14-01553-f002:**
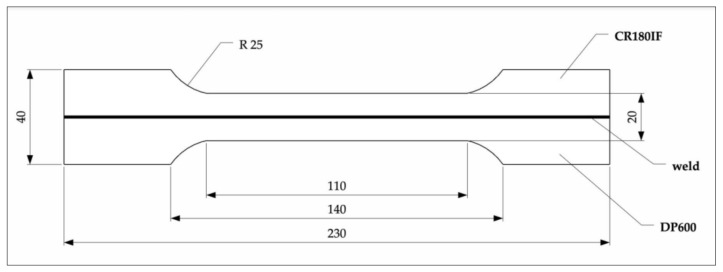
Specimens for tensile test, laser-welded specimen (unit: mm).

**Figure 3 materials-14-01553-f003:**
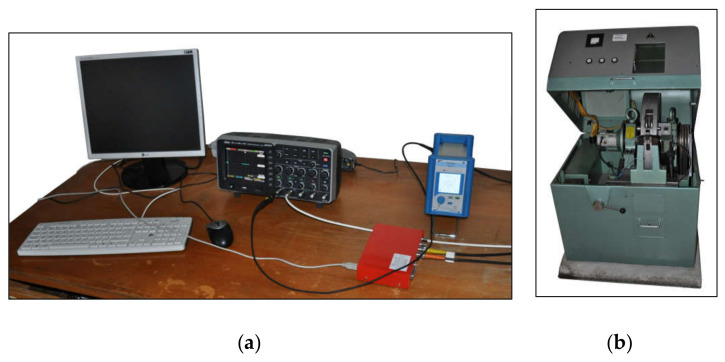
Equipment for dynamic test: (**a**) LeCroy Wave Ace oscilloscope and Kistler 5015A charge amplifier; (**b**) rotary hammer RSO.

**Figure 4 materials-14-01553-f004:**
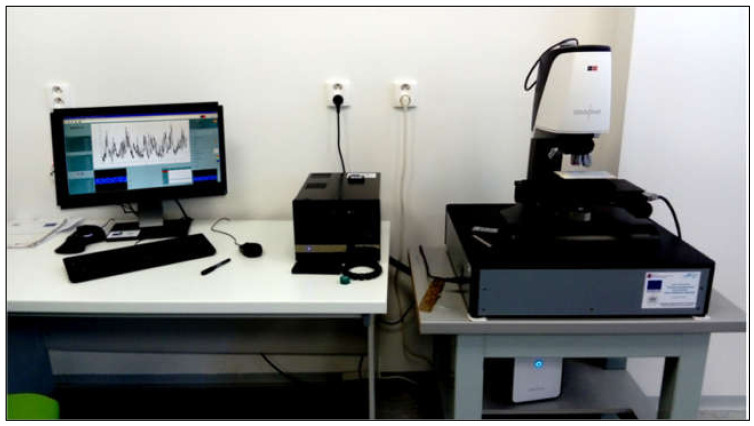
Optical profiler PLu neox 3D for surface topography measurements.

**Figure 5 materials-14-01553-f005:**
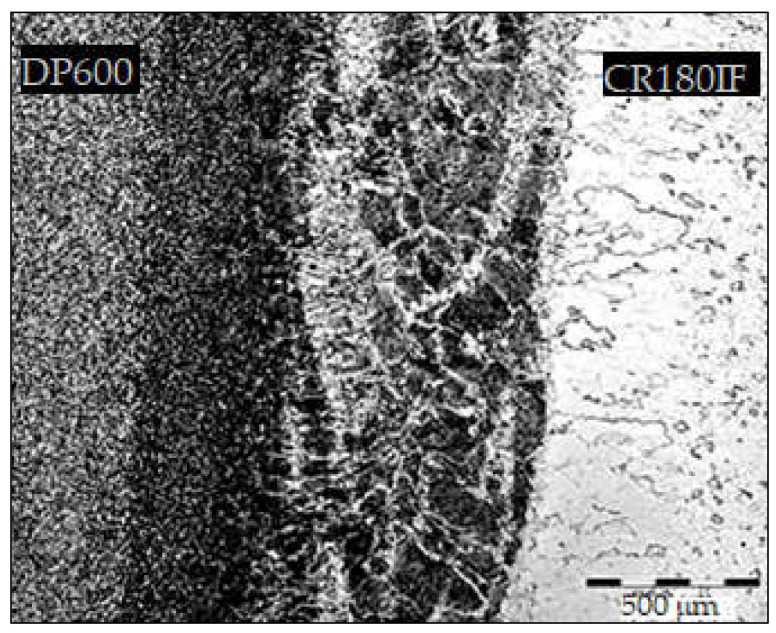
The microstructure of welded joint CR180IF and DP600.

**Figure 6 materials-14-01553-f006:**
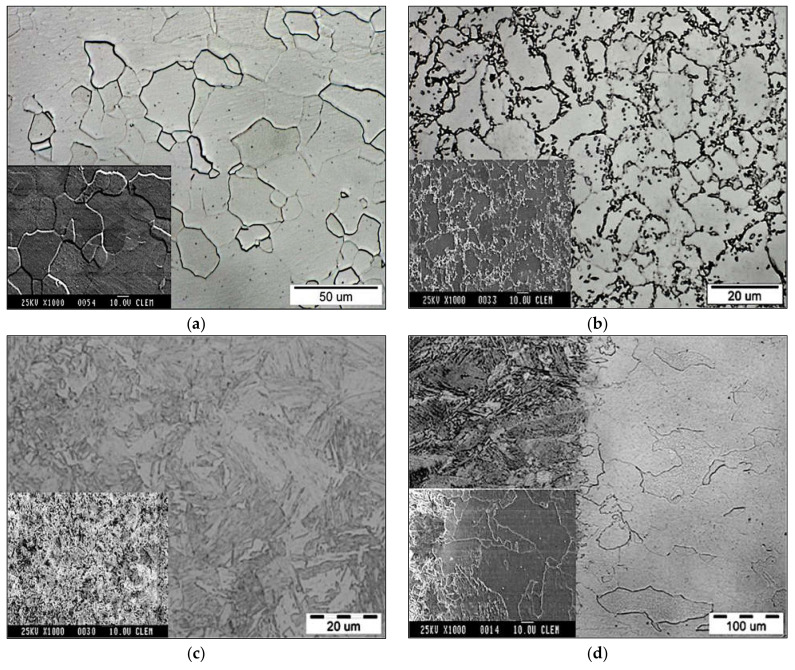
The microstructure of investigated steels: (**a**) CR180IF steel; (**b**) DP600 steel; (**c**) fusion zone; (**d**) heat-affected zone of CR180IF.

**Figure 7 materials-14-01553-f007:**
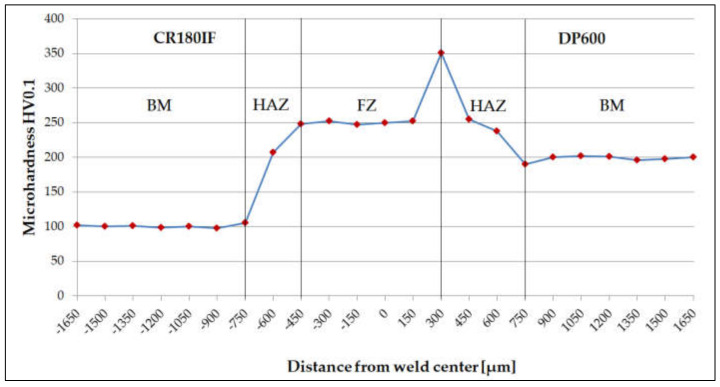
The microhardness distribution in base metal (BM), heat-affected zone (HAZ), and welded fusion zone (FZ).

**Figure 8 materials-14-01553-f008:**
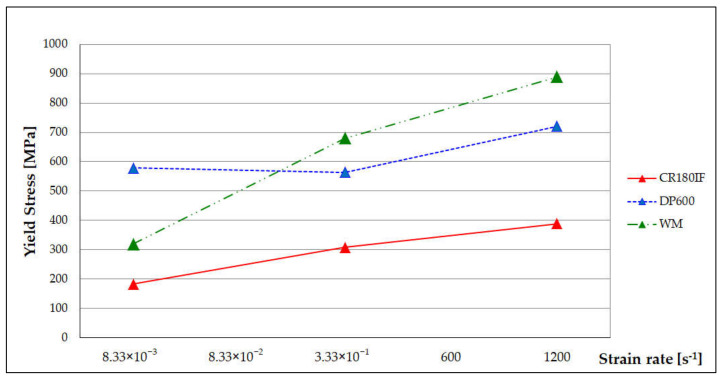
Dependence of the yield strength (YS) on the tested steels’ strain rate and welded material.

**Figure 9 materials-14-01553-f009:**
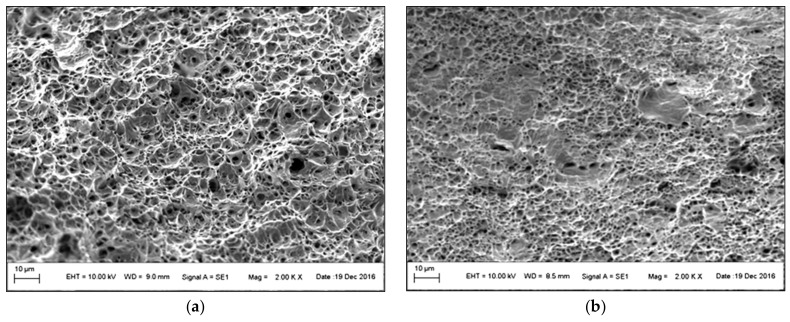
Fracture surface of CR180IF steel: (**a**) under static loading 8.33 × 10^−3^ s^−1^, (**b**) under dynamic loading 1200 s^−1^.

**Figure 10 materials-14-01553-f010:**
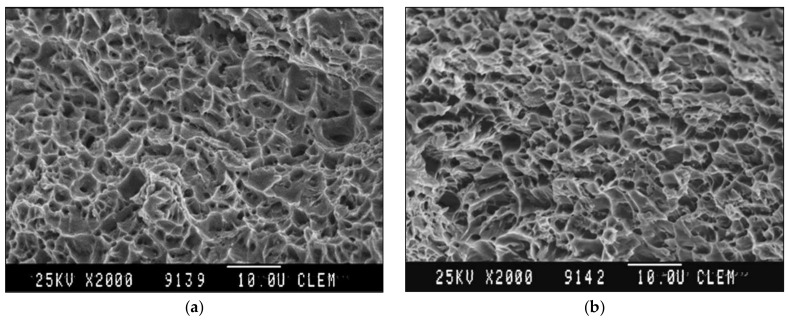
Fracture surface of DP600 steel: (**a**) under static loading 8.33 × 10^−3^ s^−1^, (**b**) under dynamic loading 1200 s^−1^.

**Figure 11 materials-14-01553-f011:**
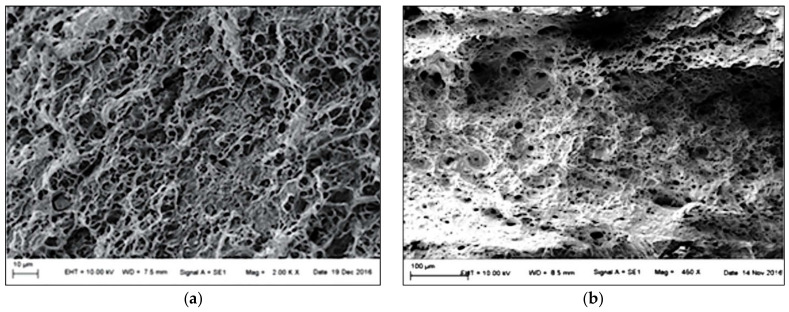
Fracture surface of the fusion zone: (**a**) under static loading 8.33 × 10^−3^ s^−1^, (**b**) under dynamic loading 1200 s^−1^.

**Figure 12 materials-14-01553-f012:**
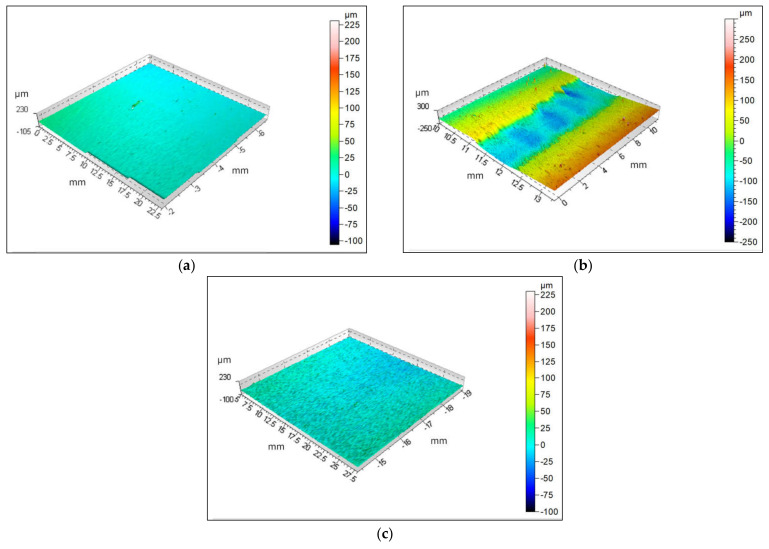
The 3D map: (**a**) base material CR180IF, (**b**) welded joint, (**c**) base material DP600.

**Table 1 materials-14-01553-t001:** Chemical compositions of investigated steels (wt %).

Steel	C	Si	Mn	P	S	Ti	Al	Nb	V	N
CR180IF	0.01	0.006	0.082	0.011	0.011	0.04	0.055	0.001	0.002	0.002
DP600	0.072	0.005	1.18	0.017	0.01	0.002	0.057	0.002	0.002	0.005

**Table 2 materials-14-01553-t002:** Parameters of laser welding by solid-state fiber laser YLR 4500.

Steel	Laser Power (W)	Welding Speed (mm.s^−1^)	Focal Length (µm)	Feeding Fiber Diameter (µm)	Collimation (mm)	Shielding Gas
CR180IF DP600	2000	40	200	100	200	Helium

**Table 3 materials-14-01553-t003:** Measured values of microhardness HV0.1.

Material	Base Metal HV0.1	Heat Affected Zone HV0.1	Fusion Zone HV0.1
CR180IF	102 ± 2	213 ± 2	253 ± 2
DP600	197 ± 2	245 ± 2	350 ± 2

**Table 4 materials-14-01553-t004:** Mechanical properties of investigates steels CR180IF, DP600, and welded material (WM).

Strain Rate (s^−1^)	Material	Yield Stress YS (MPa)	Ultimate Tensile Strength UTS (MPa)	Elongation A (%)	YS/UTS
8.33 × 10^−3^	CR180IF	186	320	35	0.58
	DP600	393	579	25.3	0.68
	WM	319	483	22.8	0.66
3.33 × 10^−1^	CR180IF	308	518	21	0.59
	DP600	564	701	23	0.80
	WM	680	832	24	0.81
1200	CR180IF	389	546	32	0.71
	DP600	721	785	31	0.79
	WM	889	1050	39	0.84

## Data Availability

Data sharing is not applicable for this article.

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
