# Peer review of "The Performance of CR180IF and DP600 Laser Welded Steel Sheets under Different Strain Rates"

_materials, 2021, doi:10.3390/ma14061553_

Round 1

Reviewer 1 Report

The current paper investigates the effect of laser welding of two types of steel that is used in car manufacturing. The aim here is to study how laser welding affects certain mechanical properties of the steel sheets. Some of the properties investigated are microhardness, microstructure and the roughness of welded joints. The aim also is to determine how these welded sheets behave under dynamic conditions for example during impact. The authors use RSO for dynamic testing and tensile tests were carried out to obtain mechanical properties under static and dynamic conditions. This procedure allowed obtaining the material data under various strain rates. The results showed that the strain rate had an influence on the mechanical properties such as the yield strength.

The abstract needs more work, it is not clear what exactly the authors have done in this study, they authors talk about some test then move and talk about another one then another one. Which is quiet confusing, the authors should try to be clearer and more concise, say what you have done and what you found.

The title does not reflect the work done in this paper, perhaps consider changing the title to the following or similar: The performance of CR180IF and DP600 Laser Welded Steel Sheets under different strain rates

Please consider reviewing the abstract and highlight the novelty, major findings, and conclusions.
What is the research gap did you find from the previous researchers in your field? Mention it properly. It will improve the strength of the article.

The authors should avoid writing small paragraphs such as in lines 87-88 and 89-90, instead combine them all in one bigger paragraph, apply this everywhere else in the manuscript

Why did the authors choose those specific parameters in Table 2, are they based on machine range or industrial recommendations or you are testing a new range of welding parameters for example? please explain and support with references if possible.

Section 3.1 replace microstructure with the word microstructural (recommended)

The average grain size is 25 microns, how did you measure that?

Add some images for the devices which you used measuring hardness and surface roughness and the measurement setup for each in the materials and methods sections

Ok now I see, in section 3.2 a lot of the information there can be moved to the materials and method sections. Line 134-142

Figure 6 needs to be moved back to the materials and methods sections

The paper needs organisation, many parts in results and discussion are more fit for the materials and methods section

Combine table 4-6 in one larger table

Again please don’t write small paragraphs combine them into bigger one the paper is very fragmented this way

There are so man SEM images but very little discussion about the findings from the study.

The results are merely described and is limited to comparing the experimental observation. The authors are encouraged to include a discussion section and critically discuss the observations from this investigation with existing literature.

Paper can not be accepted in its current form

Author Response

Dear Reviewer,

Thank you for giving me the opportunity to submit a revised draft of manuscript titled: “Examination of Mechanical Properties and Surface Roughness of CR180IF and DP600 Laser Welded Steel Sheets” after the change of Title to: “The Performance of CR180IF and DP600 Laser Welded Steel Sheets under Different Strain Rates.

Please find attached the point-by-point response to the Reviewer’s comments.

We are grateful to the Reviewer for their insightful comments on our paper.

Sincerely,

Kristína Zgodavová

Reviewer 2 Report

Dear Authors,

The reviewer paper is an experimental work titled: "Examination of Mechanical Properties and Surface Roughness of CR180IF and DP600 Laser Welded Steel Sheets". The work describes the preparation, implementation and research results of dissimilar joints from two grades of steel used in the automotive industry welded with a laser. The subject matter is of great practical importance, however I have some important comments. I present them below in the order of reading the manuscript.

I propose to remove: “Examination of” from title.

Please arrange the abstract according to the scheme: introductory sentence, aim, the list of research, research results. Please complete the abstract with the most important quantitative results.

Sentence: "The laser welds microstructure and microhardness were evaluated in the base metal, heat-affected zone, and fusion zone." It should be: “The laser welded joints microstructure and microhardness were evaluated”. Please pay attention to the meaning of the terms “weld” and “welded joint”.

Keywords: please add: "dissimilar" before "laser welding".

Introduction:

Lines 36,37: The phrase "stronger, lighter, and stiffer" doesn't sound technical. Please edit them.

Line 43: please list these traditional processes.

Lines 52,53: This sentence is redundant.

Line 61: explain the entire abbreviation (add: advanced).

Lines 73, 74: This description is true only for autogenous welding, and filler metal is often used.

Line 76: change: "amount of heat introduced" to: "heat input". "Weld" or "welded joint"?

I propose to extend this chapter: there is no information about the disadvantages of laser welding, the advantage of this process is the ability to perform dissimilar joints (and this is the subject of the manuscript). Additionally, I suggest that one paragraph be devoted to the problems of weldability of dissimilar joints resulting from, for example, differences in the physico-chemical properties of the joined materials. Among the current works recently published in mdpi, I can recommend the following articles: https://doi.org/10.3390/ma13204540, https://doi.org/10.3390/ma13132930, https://doi.org/10.3390/met10091138

Lines 80-85: The article has the standard IMRaD structure, so this description can be removed.

Chapter 2:

Line 87: change: "observed" to "investigated" or "tested".

What were the dimensions of the welded sheets? Were they fixed on the welding table during welding? This is important as it affects the welding thermal cycle.

Why was He used? How was his purity? What was the flowrate of shielding gas?

The journal's guidelines recommend providing the names and addresses of the manufacturers of all devices used in the research.

Lines 100-101: This sentence can be deleted.

Line 108: "extension"? Maybe elongation?

Results and Discussion

Lines 134-140: this information should be given in advance in M and M chapter.

Line 150: change: "shape" to: "distribution".

Carbon equivalent has the CE or CEIIW (International Institute of Welding) symbol standard.

Line 165: "bar"? In the case of thin sheets, this word is not correct. Maybe it's better to use "specimen".

Lines 167-171: this should be transferred to M and M chapter.

Figure 7 contains the same data as tables 4-6, so it is not needed.

Conclusions chapter should be titled: "Conclusions".

Please number the conclusions.

Conclusion 1: "created" doesn't sound technical.

The first sentence of conclusion 2 can be deleted and the rest can be combined into one and the hardness results can be quantified.

Conclusion 6 is not well written: grammar.

Conclusion 7: delete the first sentence.

The last sentence of the article about future research is not needed.

References should be formatted (abbreviations of journal titles).

Author Response

(The authors gave the same response as above.)

Round 2

Reviewer 1 Report

authors have answered all questions 

Reviewer 2 Report

Dear Authors,
thank you very much for making corrections according to my suggestions and all the answers. I believe that the work can be published after authors proofreading. Pay attention to the notation of citations, e.g. [31-33] and authors' names, e.g. references: 21, 22.

This manuscript is a resubmission of an earlier submission. The following is a list of the peer review reports and author responses from that submission.